🔓 | Open Peer Review | Bacteriology | Research Article

# *Coxiella burnetii* and *Bartonella* species serology of febrile patients with an established infectious or inflammatory diagnosis in Sudan, Nepal, and Cambodia

Carl Boodman,[1,2,3] Sophie Edouard,[4,5] Johan van Griensven,[2] Kanika Deshpande Koirala,[6] Basudha Khanal,[6] Suman Rijal,[6] Narayan Raj Bhattarai,[6] Sayda El Safi,[7] Thong Phe,[8] Kruy Lim,[8] Pascal Lutumba,[9] François Chappuis,[10] Cédric P. Yansouni,[11,12] Achilleas Tsoumanis,[2] Barbara Barbé,[2] Marjan van Esbroeck,[2] Kristien Verdonck,[2] Marleen Boelaert,[2] Nitin Gupta,[13] Pierre-Édouard Fournier,[4,5] Emmanuel Bottieau[2]

**ABSTRACT** *Coxiella burnetii* and *Bartonella* species cause febrile illness and infective endocarditis in low- and middle-income countries (LMICs). This study investigated whether seropositivity to *C. burnetii* or *Bartonella* could be detected among patients with persistent fever for which an infectious or inflammatory etiological diagnosis had been previously established in three LMICs. Our study tested sera from Cambodian, Nepalese, and Sudanese participants using indirect immunofluorescent antibody assays (IFA) for *C. burnetii* and *Bartonella*. Seropositivity rates for both pathogens were assessed across tropical and inflammatory etiologies of fever and compared to ubiquitous bacterial infections considered as a "reference group," as they were not expected to cause serologic cross-reactivity. A total of 1,313 individuals underwent IFA, including 560/1,313 (42.7%) from Sudan, 432 (32.9%) from Nepal, and 321 (24.5%) from Cambodia. Overall, 57 (4.3%) and 60 (4.6%) participants tested positive for *C. burnetii* and *Bartonella* species, respectively. Forty-four (3.4%) individuals tested positive for both *C. burnetii* and *Bartonella* species (75.4% positive agreement). *C. burnetii* positivity did not differ significantly between the three countries ($P = 0.44$), while *Bartonella* seropositivity was predominantly identified in Nepal ($P < 0.001$). Compared to the reference group, *C. burnetii* and *Bartonella* seropositivity were more common among participants with visceral leishmaniasis, *P. falciparum* malaria, leptospirosis, brucellosis, scrub typhus, and systemic lupus erythematosus (SLE), though only statistically significant for the latter two diagnoses. Further studies are necessary to investigate *C. burnetii* and *Bartonella* seropositivity in LMICs and to disentangle cross-reactivity, previous infection, or co-infection.

**IMPORTANCE** *Coxiella burnetii* and *Bartonella* spp. are important but under-recognized causes of febrile illness and infective endocarditis in low- and middle-income countries (LMICs). This study evaluated the seroprevalence of these pathogens among patients with confirmed causes of persistent fever in Sudan, Nepal, and Cambodia. Despite the diagnostic utility of serologic testing for these infections, its performance in LMICs —where co-infections and background seropositivity are common—remains poorly characterized. The findings suggest notable seropositivity for *C. burnetii* and *Bartonella* among patients with a set of tropical and inflammatory diagnoses, including visceral leishmaniasis, *Plasmodium falciparum* malaria, leptospirosis, brucellosis, scrub typhus, and systemic lupus erythematosus. These results highlight the potential for cross-reactivity and underscore the need for context-specific validation. Enhanced understanding of serologic test characteristics is essential for accurate diagnosis in resource-limited settings.

Address correspondence to Carl Boodman, cboodman@itg.be, boodmanc@myumanitoba.ca.

Marleen Boelaert passed away during the preparation of this manuscript.

The authors declare no conflict of interest.

See the funding table on p. 10.

**KEYWORDS** *Coxiella*, *Bartonella*, febrile illness, endocarditis, serology

Infections due to zoonotic and vector-borne diseases, including those caused by *Coxiella burnetii* and *Bartonella* species, are common in low- and middle-income countries (LMICs) (1–4). Both bacteria cause febrile illness and infective endocarditis across low-resource areas globally (1–3, 5, 6). Transmission of *C. burnetii,* the etiologic agent of Q fever, predominantly occurs via inhaling aerosols contaminated by the excreta and birth products of infected animals, though transmission via contaminated animal product ingestion may also occur (2). Over 40 different *Bartonella* species infect a wide range of animal reservoirs (7). *Bartonella* species are primarily transmitted via hematophagous arthropod vectors, including lice, fleas, ticks, and sand flies (7, 8). The most common *Bartonella* species to infect humans are the louse-borne *B. quintana* (historically, the agent of trench fever), the flea-borne *B. henselae* (cat scratch disease), and the sandfly-borne *B. bacilliformis* (Carrion's disease/Oroya fever), though human infections due to other *Bartonella* species have been described (7, 9).

*C. burnetii* and *Bartonella* species are fastidious bacilli that cannot be identified by routine culture on agar with 5-day incubation (2, 7). These pathogens are thus considered culture-negative bacteria, necessitating specialized microbiologic techniques, including serologic and molecular methods for identification (2, 9). Symptoms due to both infections are frequently nonspecific, which further complicates diagnosis (2, 7). In light of these considerations, the revised modified Duke criteria include seropositivity to *C. burnetii* and *Bartonella* species with immunoglobulin G (IgG) titers ≥ 1:800 as major criteria for diagnosing infective endocarditis (10, 11).

Serologic testing for *C. burnetii* and *Bartonella* is mainly performed via indirect immunofluorescence antibody assays (IFA), which are limited by low specificity. Cross-reactivity occurs between different species within the *Bartonella* genus and between *Bartonella* species and *C. burnetii* (12–14). Furthermore, *C. burnetii* IFA cross-reacts with *Legionella*, *Chlamydia*, *Ehrlichia*, *Anaplasma*, and *Rickettsia* serologies (15–19).

Despite the central role of serology in diagnosing *C. burnetii* and *Bartonella* infections and the prevalence of these diseases in LMICs, few studies evaluate the performance of *C. burnetii* and *Bartonella* serologic testing in LMICs where other infections are common. A thorough understanding of *Coxiella burnetii* and *Bartonella* serologic test characteristics is crucial for accurately diagnosing infection and interpreting seroprevalence studies in low-resource settings.

The NIDIAG consortium ("Better DIAGnosis of Neglected Infectious Diseases," NIDIAG) was established in 2010 to address knowledge gaps in infectious syndromes in LMICs, including persistent fever in Sudan, Nepal, Cambodia, and the Democratic Republic of the Congo (DRC) (20). This prospective multicentric study focused on severe infections known to cause persistent fever (e.g., malaria, tuberculosis, visceral leishmaniasis, enteric fever, brucellosis, melioidosis, leptospirosis) but also included other infectious syndromes (e.g., community-acquired pneumonia, skin and soft tissue infections, bacterial meningitis, cholangitis) (20, 21). NIDIAG samples were analyzed in national and international (France, Belgium) reference laboratories, and the remaining samples were biobanked in French and Belgian laboratories (21). Testing for *Coxiella burnetii* and *Bartonella* species was not included in the original NIDIAG protocol (21). Our study aimed to assess *Coxiella burnetii* and *Bartonella* seropositivity among biobanked serum samples from the NIDIAG fever cohort with existing diagnoses as determined by the original NIDIAG fever studies (20, 21).

## MATERIALS AND METHODS

### Study design and populations

The original NIDIAG fever samples were collected from January 2013 to October 2014 from the following study sites in Sudan, Nepal, Cambodia, and the DRC: a rural hospital in Gedaref State, Sudan; a rural district hospital and an outpatient health center in

Mosango, Kwilu province, DRC; a rural district hospital in Dhankuta and a university hospital in Dharan, Nepal; and a referral-level Sihanouk hospital center of Hope in Phnom Penh, Cambodia (20). The NIDIAG fever study compiled clinical and diagnostic data from consenting patients over 5 years of age presenting with over 7 days of fever (21). Patients requiring initial intensive care were excluded (21). Insufficient volume of serum originating from the DRC precluded analysis; the included samples originated from Nepal, Cambodia, and Sudan. All samples of patients in whom an infectious or inflammatory condition had been diagnosed during the NIDIAG studies were included, provided that sufficient serum was available for testing. Original NIDIAG diagnoses included the following: ubiquitous culture-positive bacterial infections (further considered as the reference group as these infections were not expected to interfere with *C. burnetii* or *Bartonella* spp. serology; see more detail below); tropical diseases such as malaria, visceral leishmaniasis, leptospirosis, etc.; and one autoimmune condition, systemic lupus erythematosus (SLE). The latter two groups were hypothesized to demonstrate serologic cross-reactivity to *C. burnetii* and/or *Bartonella* species. No samples were previously tested for *C. burnetii* and *Bartonella* species in the initial studies.

Due to the persistence of immunoglobulin G (IgG) to *C. burnetii* and *Bartonella* antigens after infection, the reference group was created as a proxy for background seroprevalence. This group comprised all participants diagnosed with ubiquitous extracellular bacterial infections likely accounting for the presenting fever, but not typically associated with the clinical features of *C. burnetii* infection or bartonellosis, and not known to cause serologic cross-reactivity with these pathogens. All participants meeting the following criteria were included in the reference group: diagnosed with a common infectious syndrome that is not typically associated with *C. burnetii* or *Bartonella* infection; associated with a cultivated bacterial pathogen with no known serologic cross-reactivity to *C. burnetii* or *Bartonella* species; and had remaining serum available for testing. The reference group included the following syndromes: bacterial meningitis, cholangitis/cholecystitis, tonsillitis/pharyngitis, skin and soft tissue infection, bacterial gastroenteritis/dysentery, bacterial liver abscess, appendicitis, spontaneous bacterial peritonitis (SBP), pyelonephritis, and documented bacteremia with non-fastidious organisms (other than *Salmonella* and *Brucella* species; cases of enteric fever and brucellosis were analyzed separately due to their facultatively intracellular localization, similar to *Bartonella* spp.). Serologic positivity to *C. burnetii* and *Bartonella* spp. in the reference group was considered to represent positivity from previous exposure rather than an association with the participant's presenting fever.

## Indirect immunofluorescent antibody assays

An in-house IFA was performed on stored serum samples to assess antibody response to *C. burnetii* antigens (phase I and II lipopolysaccharides) and *Bartonella* antigens (*B. quintana* and *B. henselae* antigens), as previously described (2, 9, 12, 22). Sera were diluted in phosphate-buffered saline with 3% powdered milk to mitigate nonspecific antibody fixation (12). Initial screening was conducted using titers of 1:50 and 1:100. Titers ≥ 1:100 were considered positive and underwent further doubling dilutions (9, 12, 23). This cut-off is the established positivity threshold for the in-house IFA assay and is considered equivalent to the 1:128 cut-off used in the FOCUS-DiaSorin IFA assay (9, 12, 23). Samples exhibiting a titer of 1:50 but testing negative at 1:100 were classified as indeterminate and were not subjected to further serial dilutions (23). For *C. burnetii,* IFA screening was performed with a combination of anti-phase I and II antigens, and positive sera subsequently underwent serial dilution and testing for anti-phase I and II IgG. For *Bartonella,* screening was performed with *B. henselae* and *B. quintana* antigens. Samples with IgG titers ≥ 1:100 to either *B. quintana* or *B. henselae* antigens were counted as a positive *Bartonella* result.

## Analysis and interpretation

Participants with *C. burnetii* anti-phase I IgG titers ≥ 1:800 or anti-*Bartonella* IgG titers ≥ 1:800 were described separately and linked to the original clinical descriptions. These thresholds were chosen for focused analysis as they are major diagnostic criteria for infective endocarditis (10). Due to known serologic cross-reactivity between *C. burnetii* and *Bartonella* species, the antibody response with the highest titer was deemed to indicate exposure to the causative genus. Separate analyses compared IFA results from the reference group with results from participants diagnosed with visceral leishmaniasis (VL), tuberculosis, malaria (*Plasmodium falciparum* and *Plasmodium vivax*), leptospirosis, melioidosis, brucellosis, scrub typhus, enteric fever, and SLE as diagnosed using the original NIDIAG fever protocol (21).

Visceral leishmaniasis (VL) was diagnosed using a combination of either promastigote culture (Novy–McNeal–Nicolle medium), microscopic visualization of Giemsa-stained amastigotes, or direct agglutination test (DAT) and response to treatment (21). TB was diagnosed using either microscopic visualization of acid-fast bacilli (Ziehl-Neelsen) and/or the GeneXpert Mtb/Rif assay. Malaria (*P. falciparum* and *P. vivax*) was diagnosed using microscopy of Giemsa-stained thick and thin films and/or rapid diagnostics tests targeting both the pan-pLDH and HRP2/3 antigens (SD Malaria Ag P.f/Pan and CareStart malaria) (21, 24, 25). Leptospirosis was diagnosed using either polymerase chain reaction (PCR) and/or microscopic agglutination tests (MAT). Melioidosis was diagnosed by a positive bacterial culture for *Burkholderia pseudomallei*. Brucellosis was diagnosed using either bacteriologic culture and/or serology. Scrub typhus was diagnosed using serology detecting immunoglobulin M (IgM) to *Orientia tsutsugamushi* antigens (21). Enteric fever was diagnosed by blood culture growth of *Salmonella enterica* serotypes Typhi and Paratyphi. SLE was diagnosed by positive antinuclear antibody test (ANA) in combination with clinical symptoms compatible with SLE, as determined by the treating clinician.

## Statistical methods

Statistics were performed using R version 4.2.2 software (2022-10-31). The chi-squared and Fisher's exact tests were used to compare categorical variables, with the latter used for smaller sample sizes. The chi-squared test was performed to compare seropositivity between the three countries to assess whether seropositivity was equal or different between the countries. Fisher's exact test was used to compare seropositivity rates between the disease and reference groups. Unless otherwise specified, comparisons were performed using positive serologies (titers ≥ 1:100). Kappa coefficient and positive and negative agreements were calculated to assess potential cross-reactivity between *C. burnetii* and *Bartonella* IFAs. Relative risk (RR) ratios were calculated to determine the probability of seropositivity in the diseased group compared to the reference group. Attributable risk percent (AR%) was calculated to estimate the percentage of serologic positivity attributable to exposure to the alternate disease. The threshold for statistical significance was set at a *P*-value of <0.05.

## RESULTS

### Overall seropositivity and country-specific seropositivity rates

A total of 1,313 individuals, including the reference group, underwent serologic testing for *C. burnetii* and *Bartonella* species. This involved 560 (42.7%) individuals from Sudan, 432 (32.9%) from Nepal, and 321 (24.5%) from Cambodia (Table 1). Overall, 57 (4.3%) individuals tested positive for *C. burnetii,* 182 (13.9%) individuals had indeterminate testing, and 1,074 (81.8%) tested negative. Seropositivity rates were 5.2% (29/560), 3.7% (16/432), and 3.7% (12/321) for Sudan, Nepal, and Cambodia, respectively (Table 1). *C. burnetii* positivity did not differ significantly between the three countries ($X^2$ [df = 2, *N* = 1,313] =1.65, *P* = 0.44). Regarding testing for *Bartonella,* 60 (4.6%) individuals were positive, 73 (5.6%) had indeterminate IFA, and 1,181 (89.9%) tested negative. *Bartonella* seropositivity rates were 1.6% (9/560), 11.3% (49/432), and 0.6% (2/321) for Sudan, Nepal,

**TABLE 1** *C. burnetii* and *Bartonella* species seropositivity among febrile individuals in Sudan, Nepal, and Cambodia according to country[a]

|  | Country | | | | P value |
|---|---|---|---|---|---|
|  | **Sudan** | **Nepal** | **Cambodia** | **Total** |  |
| N positive *C. burnetii* IFA (seroprev) | 29 (5.2%) | 16 (3.7%) | 12 (3.7%) | 57 (4.3%) | 0.44 |
| N positive *Bartonella* IFA (seroprev) | 9 (1.6%) | 49 (11.3%) | 2 (0.6%) | 60 (4.6%) | <0.001 |
| Total | 560 | 432 | 321 | 1,313 |  |

[a]N positive *C. burnetii* indirect immunofluorescent antibody assay (IFA, seroprev): number of participants with positive *C. burnetii* IFA and associated seroprevalence rate from the affiliated country. N positive *Bartonella* IFA (seroprev): number of participants with positive Bartonella spp. IFA with the associated seroprevalence rate from the affiliated country. *P*-value: *P*-value of chi-squared statistic comparing seropositivity rates between the three countries.

and Cambodia, respectively. *Bartonella* seropositivity differed significantly between the three countries ($X^2$ [2, $N = 1{,}313$ ]=68.18, $P < 0.001$).

## Agreement between *C. burnetii* and *Bartonella* IFA

There were 1,024 (78.0%) agreements between *C. burnetii* and *Bartonella* IFA results (kappa coefficient = 0.14, 95% CI: 0.08–0.19), indicating slight agreement. Percent positive and negative agreements were 75.4% (95% CI: 93.0%–77.7%) and 91.7% (95% CI: 90.1%–93.1%), respectively. Forty-four individuals tested positive for both *C. burnetii* and *Bartonella* species. Agreement increased when indeterminate results were considered positive, with a resulting kappa coefficient of 0.41 (95% CI: 0.32–0.50).

## Reference group

A total of 67 cases met criteria to be included in the reference group, including 14 (20.9%) cases of skin and soft tissue infection, 13 (19.4%) tonsillitis/pharyngitis, 10 (14.9%) bacteremia, 10 (14.9%) bacterial gastroenteritis/dysentery, 6 (9.0%) bacterial meningitis, 5 (7.5%) spontaneous bacterial peritonitis, 4 (6.0%) cholangitis/cholecystitis, 2 (3.0%) cases of appendicitis, 2 (3.0%) bacterial liver abscesses, and 1 (1.5%) pyelonephritis. Most cases (37/67 = 55.2%) were from Cambodia. 27 (40.3%) were from Sudan and 3 (4.5%) from Nepal. Two (3.0%) individuals in the reference group screened positive for *C. burnetii,* and 9 (13.4%) had indeterminate serology. One individual (1.5%) screened positive for *Bartonella* species, and one individual (1.5%) was *Bartonella* indeterminate.

## Visceral leishmaniasis

One hundred nine participants were diagnosed with visceral leishmaniasis. Of these 109 participants, 6 (5.5%) tested positive for *C. burnetii*, 15 (13.8%) were indeterminate, and 88 (80.7%) tested negative (Table 2). *C. burnetii* seropositivity was more common among individuals with VL than in the reference group (RR = 1.844, 95% CI: 0.38–88.87), though this difference was not statistically significant ($P = 0.47$). Ten (9.2%) and eight (7.3%) participants with VL tested positive and indeterminate for *Bartonella* IFA, respectively (Table 3). *Bartonella* seropositivity was more common among individuals with VL than in the control group (RR = 6.15, 95% CI: 0.81–46.94). This difference tends toward statistical significance ($P = 0.051$). Two participants diagnosed with VL had *C. burnetii* IFA titers > 1:800 (Table 4).

## Tuberculosis

Ninety-three participants were diagnosed with TB. Two (2.2%) tested positive for *C. burnetii*, and 7 (7.5%) had indeterminate serologies (Table 2). *C. burnetii* seropositivity rate was similar among individuals diagnosed with TB and those in the reference group ($P = 1$). No participants with TB tested positive for *Bartonella* IFA, though four (4.3%) had indeterminate *Bartonella* serology (Table 3). One Nepalese participant diagnosed with TB had *C. burnetii* anti-phase I IgG titers of 1:800 and anti-phase II IgG titers of 1:200 (Table 4).

**TABLE 2** *C. burnetii* seropositivity by disease and compared to the reference group[a]

| | *C. burnetii* seroposi-tive/total tested (%) | *C. burnetii* indetermi-nate/total tested (%) | Relative risk ratio (95% CI) | Attributable risk percent (95% CI) | *P* value |
|---|---|---|---|---|---|
| Reference group | 2/67 (3.0%) | 9/67 (13.4%) | N/A[b] | N/A | N/A |
| VL | 6/109 (5.5%) | 15/109 (13.8%) | 1.84 (0.4–88.9) | 45.8% (−69.4 to 160.9) | 0.47 |
| TB | 2/93 (2.2%) | 7/93 (7.5%) | 0.78 (0.1–5.4) | −28.36 (−247.5 to 190.8%) | 1 |
| *P.f* malaria | 2/46 (4.3%) | 17/46 (37.0%) | 2.09 (0.3–14.2) | 53.2% (80.4–184.9) | 0.59 |
| Leptospirosis | 5/52 (9.6%) | 5/52 (9.6%) | 3.56 (0.7–17.6) | 71.94 (12.2%–156.1%) | 0.12 |
| Melioidosis | 0/13 (0.0%) | 1/13 (7.7%) | 0 | N/A | 1 |
| Brucellosis | 2/18 (11.1%) | 5/18 (27.8%) | 3.722 (0.6–24.7) | 73.13 (−26.0 to 172.3%) | 0.12 |
| Scrub typhus | 0/11 (0.0%) | 0/11 (0.0%) | 0 | N/A | 1 |
| Enteric fever | 2/22 (9.1%) | 0/22 (0.0%) | 3.04 (0.5–20.4) | AR%: 67.2 (−42.6% to 176.9%) | 0.25 |
| SLE | 3/7 (42.9%) | 0/7 (0.0%) | 14.36 (2.9–71.9) | 93.03 (47.4%–138.6%) | **0.005**[c] |

[a]*C. burnetii* seropositive/total tested (%): number of participants who tested positive for *C. burnetii* (Cb) (titers ≥ 1:100) over total, with percent positivity in parentheses. *C. burnetii* indeterminant/total tested (%): number of participants who tested indeterminant for *C. burnetii* (titers = 1:50) over total with percent positivity. Relative risk ratio (95% CI): relative risk ratio of positive cases from associated disease and control with 95% CI (CI). Attributable risk percent (95% CI): attributable risk percent with 95% CI. *P* value: Fisher's exact test *P* value. VL, visceral leishmaniasis; *P. f* malaria, *Plasmodium falciparum* malaria; SLE, systemic lupus erythematosus.
[b]N/A, not applicable.
[c]Bold value indicates statistical significance.

## Malaria

Fifty participants were diagnosed with malaria. There were 46 (92.0%) cases of *P. falciparum* and 4 (8.0%) cases of *P. vivax*. All cases of *P. vivax* were negative for both *C. burnetii* and *Bartonella* serology, except for one participant with indeterminate *C. burnetii* serology (Tables 2 and 3). Of the 46 cases of *P. falciparum* malaria, 2 (4.3%) tested positive for *C. burnetii,* and 17 (37.0%) were indeterminate. *C. burnetii* seropositivity was more common among individuals with malaria than in the control group (RR = 2.09, 95% CI: 0.309–14.20, AR% = 53.24%, 95% CI: 80.44–184.93), though this difference was not statistically significant (*P* = 0.59). The difference became significant when comparing indeterminate results (*P* = 0.008). Furthermore, when indeterminate results were treated as positive, statistical significance remained (*P* = 0.004). One (2.2%) participant with *P. falciparum* malaria tested positive for *Bartonella,* and three (6.5%) had indeterminate *Bartonella* IFA. *Bartonella* seropositivity was equally common among individuals with malaria and the control group (*P* = 1).

**TABLE 3** *Bartonella* seropositivity by disease compared to the reference group[a,c]

| | *Bartonella* seroposi-tive/total tested (%) | *Bartonella* indetermi-nant/total tested (%) | Relative risk ratio (95% CI) | Attributable risk percent (95% CI) | *P* value |
|---|---|---|---|---|---|
| Reference group | 1/67 (1.5%) | 1/67 (1.5%) | N/A[b] | N/A | N/A |
| VL | 10/109 (9.2%) | 9/109 (8.3%) | 6.15 (0.805–46.94) | 83.73% (3.45%–164.01%) | 0.05 |
| TB | 0/93 (0.0%) | 4/93 (4.3%) | 0 | N/A | 0.45 |
| P.f malaria | 1/46 (2.2%) | 3/46 (6.5%) | 1.42 (0.09–22.2) | 29.85% (−200.2% to 259.95%) | 1 |
| Leptospirosis | 2/52 (3.8%) | 1/52 (1.9%) | 2.62 (0.244–28.18) | 61.94 (−84.26 to 208.1365) | 0.58 |
| Melioidosis | 0/13 (0.0%) | 1/13 (7.7%) | N/A | N/A | 1 |
| Brucellosis | 1/18 (5.5%) | 1/18 (5.5%) | 3.722 (0.24–56.648) | 73.13% (−68.83% to 215.03%) | 0.28 |
| Scrub typhus | 7/11 (63.6%) | 2/11 (18.5%) | 42.63 (5.794–313.76) | 97.65 (67.26%–128.05%) | **0.001** |
| Enteric fever | 0/22 (0.0%) | 1/22 (4.5%) | 0 | N/A | 1 |
| SLE | 2/7 (28.7%) | 1/7 (14.3%) | 19.14 (1.97–185.42) | 94.78 (41.03%–148.5%) | **0.02** |

[a]*Bartonella* Seropositive/total tested (%): number of participants who tested positive for *Bartonella* species (titers ≥ 1:100) over total testing, with percent positivity. *Bartonella* indeterminant/total tested (%): number of participants who tested indeterminant for *Bartonella* (titers = 1:50) over total with percent positivity. Relative risk ratio (95% CI): relative risk ratio of positive cases from associated disease and control with 95% CI. Attributable risk percent (95% CI): Attributable risk percent with 95% CI. *P* value: Fisher's exact test *P* value. VL, visceral leishmaniasis; *P. f* malaria, *Plasmodium falciparum* malaria; SLE, systemic lupus erythematosus.
[b]N/A, not applicable.
[c]Bold values indicate statistical significance.

**TABLE 4** Participants with *C. burnetii* anti-phase I IgG titers ≥ 1:800 or *Bartonella* IgG titers ≥ 1:800[a]

| Case number | Age (sex, occupation) | Original diagnosis (diagnostic modality) | Country | *C. burnetii* or *Bartonella* IFA result | Cardiopulmonary auscultation | Death |
|---|---|---|---|---|---|---|
| 1 | 28 (M, farmer) | VL (Culture, DAT, and microscopy) | Nepal | *C. burnetii* anti-phase I IgG 1:800 and anti-phase II IgG titers 1:400 | Normal examination (no murmur) | No |
| 2 | 47 (F, unspecified worker) | VL (Culture and microscopy) | Sudan | *C. burnetii* anti-phase I IgG titers of 1:1,600 and anti-phase II IgG 1:800 | Normal examination (no murmur) | No |
| 3 | 78 (M, dependent) | TB (sputum GeneXpert) | Nepal | *C. burnetii* anti-phase I IgG titers 1:800 and anti-phase II IgG titers 1:200 | Normal examination (no murmur) | No |
| 4 | 44 (M, farmer) | Leptospirosis (MAT) | Sudan | *C. burnetii* anti-phase I IgG titers of 1:3,200 and anti-phase II IgG titers of 1:1,600 | Normal examination (no murmur) | No |
| 5 | 35 (F, housewife) | Brucellosis (positive *Brucella* species blood cultures) | Sudan | *B. quintana* 1:800 and *B. henselae* 1:200 | Normal examination (no murmur) | No |

[a]M, male; F, female; VL, visceral leishmaniasis; TB, tuberculosis; DAT, direct agglutination test; MAT, microscopic agglutination test; Death, determined at 1-month follow-up.

## Leptospirosis

Fifty-two participants were diagnosed with leptospirosis. Of these 52 participants, 5 (9.6%) tested positive for *C. burnetii* and 5 (9.6%) were indeterminate (Table 2). *C. burnetii* seropositivity was more common among individuals with leptospirosis than in the control group (RR = 3.56383, 95% CI: 0.72–17.59), though not statistically significant ($P = 0.12$). Two (3.8%) participants with leptospirosis tested positive for *Bartonella,* and one (1.9%) had indeterminate *Bartonella* IFA (Table 3). One participant from Sudan had *C. burnetii* anti-phase I IgG titers of 1:3,200 and anti-phase II IgG titers of 1:1,600 (Table 4).

## Melioidosis

Thirteen participants were diagnosed with melioidosis; none tested positive for *C. burnetii* nor *Bartonella* (Tables 2 and 3). One individual (7.7%) had an indeterminate IFA for *C. burnetii,* and another had an indeterminate IFA for *Bartonella* species.

## Brucellosis

Eighteen participants were diagnosed with brucellosis. Two (11.1%) tested positive for *C. burnetii,* and 5 (27.8%) were *C. burnetii* indeterminate (Table 2). *C. burnetii* seropositivity was more common among individuals with brucellosis than in the control group (RR = 3.722, 95% CI: 0.56–24.698), though this difference was not statistically significant ($P = 0.12$). One (5.5%) participant with brucellosis tested positive for *Bartonella,* and one (5.5%) had an indeterminate *Bartonella* IFA (Table 3). One participant from Sudan with positive *Brucella* species blood cultures had anti-*Bartonella quintana* IgG titers of 1:800 (Table 4).

## Scrub typhus

Eleven participants were diagnosed with scrub typhus. No participants diagnosed with scrub typhus had positive or indeterminate *C. burnetii* serologies. However, the majority of participants diagnosed with scrub typhus (7/11 = 63.6%) had positive *Bartonella* IFAs, all of which had low positive titers of 1:100. *Bartonella* seropositivity was significantly more elevated among individuals diagnosed with scrub typhus than the reference group ($P < 0.001$, RR = 42.63, 95% CI: 5.794–313.76).

## Enteric fever

Twenty-two participants diagnosed with enteric (typhoid and paratyphoid) fever had *C. burnetii* and *Bartonella* IFA testing. Two (9.1%) participants had positive *C. burnetii* serology (Table 2). *C. burnetii* seropositivity was similar among individuals diagnosed

with enteric compared to the control group (*P* = 0.25). One (4.5%) individual diagnosed with enteric fever had indeterminate *Bartonella* IFA results (Table 3). None had positive *Bartonella* serology.

## Systemic lupus erythematosus

Seven participants were diagnosed with SLE. Three (42.9%) participants had positive *C. burnetii* serology (Table 2). *C. burnetii* seropositivity was more common among individuals diagnosed with SLE compared to the control group (*P* = 0.005, RR = 14.36, 95% CI: 2.86–71.89). Two (28.7%) individuals diagnosed with SLE had positive *Bartonella* IFA results (Table 3). *Bartonella* seropositivity was more common among individuals diagnosed with SLE compared to the control group (*P* = 0.017, RR = 19.14, 95% CI: 1.97–185.42).

## DISCUSSION

Our findings suggest that seropositivity to *C. burnetii* and *Bartonella* species may be commonly identified among patients with persistent fever in Sudan, Nepal, and Cambodia. Seropositivity for either genus may be more frequently detected among cases with certain tropical or autoimmune etiologies, compared to those diagnosed with ubiquitous bacterial syndromes, though further studies with larger sample sizes are required to confirm this observation.

Seropositivity rates in our study exceed those described in most high-income countries (2, 12, 26). There was no statistically significant difference in *C. burnetii* seropositivity between Sudan, Nepal, and Cambodia. Most participants with *C. burnetii* seropositivity originated from Sudan, a country where *C. burnetii* infection is prevalent in livestock and human cases have been described (1, 27). *Bartonella* seropositivity was significantly more common among Nepalese participants. This finding may be linked to the endemicity of *B. quintana* in Nepal (28). Lice in Nepal commonly harbor *B. quintana,* and previous cases of *B. quintana* endocarditis have been described among Nepalese individuals (3, 28, 29). The association between altitude elevation and increased rates of *B. quintana* infection has been documented elsewhere and remains ecologically plausible due to body lice's niche in layers of clothing and the sartorial requirements of high-altitude settings (30).

Among 60 *Bartonella*-seropositive individuals, 44 were also seropositive for *C. burnetii*, suggesting cross-reactivity. While cross-reactivity between *C. burnetii* and *Bartonella* IFAs is well established, our study indicates possible cross-reactivity of these assays to other infections in LMICs (12). *C. burnetii* IFA may be falsely positive among individuals with brucellosis, leptospirosis, and SLE and demonstrate falsely indeterminate results among individuals with *P. falciparum* malaria. *Bartonella* IFA may be falsely positive among individuals with visceral leishmaniasis, scrub typhus, brucellosis, and SLE. Neither *C. burnetii* nor *Bartonella* seropositivity was associated with TB, melioidosis, or enteric fever. However, without cross-adsorption, protein immunoblotting, and molecular methods, our study could not distinguish serologic cross-reactivity from possible co-infection. Identifying the true cause of serological positivity is further complicated by the low sensitivity of molecular testing for *C. burnetii* and *Bartonella* species on peripheral blood specimens (2, 9, 31). Even if our study included PCR testing of paired blood samples, negative PCR results would be incapable of ruling out either disease (2, 9, 31).

The issue of serologic cross-reactivity may lead to misdiagnoses. In Tanzania, cases of *C. burnetii* and *B. quintana* disease confirmed by 16S rRNA metagenomic sequencing were initially misdiagnosed as "probable acute leptospirosis" due to positive MAT (32). Without confirmatory testing, it remains uncertain whether the NIDIAG cases of fever diagnosed as leptospirosis and scrub typhus by serology alone were infections due to these pathogens or whether *C. burnetii, Bartonella* species, or another undetermined etiology caused the fever. This highlights the need for improved diagnostic testing for culture-negative bacterial pathogens, especially in LMICs. A similar phenomenon may be at play for the NIDIAG participants diagnosed with probable SLE due to non-specific

symptoms and a positive ANA. Both *Bartonella* and *C. burnetii* are known to cause auto-antibody production, which calls into question the veracity of these auto-immune diagnoses (33–35).

While seropositivity was present at low titers in most cases, a few cases demonstrated *C. burnetii* and *Bartonella* IFA positivity with elevated titers that would meet diagnostic criteria for infective endocarditis (10). Two cases of VL, one case of leptospirosis, and one case of TB had *C. burnetii* anti-phase I IgG ≥1:800. One case of culture-proven brucellosis had *Bartonella* titers of 1:800. While culture-negative bacterial pathogens disproportionately cause infective endocarditis in LMICs, co-infection of two unusual infections in a single individual warrants further interrogation. None of the cases with elevated serologies had abnormal cardiac examinations suggestive of underlying endocarditis, though this alone does not exclude true infection. Further prospective infective endocarditis studies are needed to validate the revised, modified Duke criteria in LMICs.

This study is subject to several limitations. Without cross-adsorption, protein immunoblot or specialized culture-based or molecular techniques, IFA positivity could indicate previous exposure or cross-reactivity to a different pathogen or co-infection. Inadequate remaining sera impeded our ability to compare paired acute and convalescent sera, as follow-up was challenging in many low-resource settings (2, 36). Certain disease comparisons had low sample sizes, and the reference group included few cases from Nepal. The limited number of individuals meeting criteria for the reference group precluded our ability to subdivide the reference group into three country-specific reference groups. This limitation impeded our ability to match disease etiology with country of origin, preventing us from addressing inter-country differences in disease-specific seropositivity.

Febrile individuals in Sudan, Nepal, and Cambodia may be commonly exposed to *C. burnetii* and *Bartonella* species. Serologic tests targeting these pathogens may cross-react with other tropical and autoimmune diseases. Further studies are necessary to improve diagnostics for culture-negative bacteria and elucidate the true burden of these infections in LMICs.

## ACKNOWLEDGMENTS

We thank the laboratory technicians at IHU-Marseille's manual serology laboratories (particularly Bernard Amphoux and Catherine Brossard) for generously facilitating the diagnostic tests involved in this study.

This study/project has been funded by a grant (2024) from the European Society of Clinical Microbiology and Infectious Diseases (Europäische Gesellschaft für klinische Mikrobiologie und Infektionskrankheiten) (ESCMID) to (C.B.). The original NIDIAG work was part of the NIDIAG European research network (Collaborative Project), supported by the European Union's Seventh Framework Program for research, technological development, and demonstration under grant agreement no. 260260. Carl Boodman's salary is supported by the University of Manitoba's Clinical Investigator Program (Canada) and the Canadian Institutes of Health Research (CIHR fellowship, MFE 194076). Carl Boodman's work on Bartonella quintana is also supported by Flanders-Québec bilateral research cooperation grant (G0AEZ24N); The Research Foundation—Flanders (FWO) and Fonds de Recherche du Québec (FRQ). C.P.Y. holds a clinician-researcher scholar career award from the Fonds de Recherche du Québec—Santé (FRQS). The funders had no role in study design, data collection and analysis, decision to publish, or preparation of the manuscript.

Carl Boodman: conceptualization, methodology, formal analysis, writing—original draft, funding acquisition, project administration. Sophie Edouard: supervision, resources, methodology, writing—review & editing. Johan van Griensven: supervision, writing—review & editing. Kanika Deshpande Koirala: investigation, resources, writing—review & editing. Basudha Khanal: resources, investigation, writing—review & editing. Suman Rijal: resources, investigation, project administration, writing—review & editing.

Narayan Raj Bhattarai: resources, investigation, writing-review & editing. Sayda El Safi: investigation, resources, project administration, writing—review & editing. Thong Phe: investigation, resources, writing—review & editing. Kruy Lim: investigation, resources, writing—review & editing. Pascal Lutumba: investigation, resources, writing—review & editing. François Chappuis: investigation, resources, writing—review & editing. Cédric P. Yansouni: supervision, investigation, funding acquisition, writing—review & editing. Barbara Barbé: methodology, investigation, project administration, resources, writing—review & editing. Achilleas Tsoumanis: investigation, project administration, resources. Marjan van Esbroeck: investigation, resources, writing—review & editing, project administration. Tine Verdonck: project administration, investigation, resources, writing—review & editing. Marleen Boelaert: investigation, resources, methodology, project administration. Nitin Gupta: investigation, methodology, writing—review & editing. Pierre-Édouard Fournier: supervision, resources, project administration, methodology, writing—review & editing. Emmanuel Bottieau: supervision, resources, project administration, writing—review & editing.

The information has not previously been presented at any meetings/conferences.

## AUTHOR AFFILIATIONS

[1]Division of Infectious Diseases, Department of Internal Medicine, University of Manitoba, Winnipeg, Manitoba, Canada

[2]Department of Clinical Sciences, Institute of Tropical Medicine, Antwerp, Belgium

[3]Department of Medical Sciences, University of Antwerp, Antwerp, Belgium

[4]Institut Hospitalo-Universitaire en Maladies infectieuses (IHU–Méditerranée Infection), Marseille, France

[5]French reference center for rickettsioses, Q fever and bartonelloses, IHU–Méditerranée Infection, Marseille, France

[6]B. P. Koirala Institute of Health Sciences, Dharan, Nepal

[7]Faculty of Medicine, University of Khartoum, Khartoum, Sudan

[8]Sihanouk Hospital Center of HOPE, Phnom Penh, Cambodia

[9]Institut National de Recherche Biomédicale (INRB), Kinshasa, Democratic Republic of Congo

[10]Division of Tropical and Humanitarian medicine, Geneva University Hospitals (HUG), Geneva, Switzerland

[11]Divisions of Infectious Diseases and Medical Microbiology, McGill University Health Centre, Montreal, Québec, Canada

[12]JD MacLean Centre for Tropical and Geographic Medicine, McGill University, Montreal, Québec, Canada

[13]Department of Infectious Disease, Kasturba Medical College, Manipal, Manipal Academy of Higher Education, Manipal, India

## AUTHOR ORCIDs

Carl Boodman http://orcid.org/0000-0001-6894-2262
Marjan van Esbroeck http://orcid.org/0000-0002-7294-6319
Nitin Gupta http://orcid.org/0000-0002-9687-2836

## FUNDING

| Funder | Grant(s) | Author(s) |
| --- | --- | --- |
| Bilateral Research Cooperation Quebec-Flanders (Fonds de Recherche du Quebec/Research Foundation-Flanders) | G0AEZ24N | Emmanuel Bottieau |
| European Society for Clinical Microbiology and Infectious diseases | 2024 | Carl Boodman |

## AUTHOR CONTRIBUTIONS

Carl Boodman, Conceptualization, Formal analysis, Methodology, Project administration, Writing – original draft, Writing – review and editing | Sophie Edouard, Methodology, Project administration, Resources, Supervision | Johan van Griensven, Project administration, Resources, Supervision | Kanika Deshpande Koirala, Methodology, Project administration, Resources | Basudha Khanal, Methodology, Project administration, Resources | Suman Rijal, Methodology, Project administration, Resources | Narayan Raj Bhattarai, Project administration, Resources | Sayda El Safi, Methodology, Project administration, Resources | Thong Phe, Methodology, Project administration, Resources | Kruy Lim, Methodology, Project administration, Resources | Pascal Lutumba, Methodology, Resources | François Chappuis, Methodology, Project administration, Resources, Supervision | Cédric P. Yansouni, Conceptualization, Project administration, Resources, Supervision | Achilleas Tsoumanis, Data curation, Methodology, Resources | Barbara Barbé, Investigation, Methodology, Project administration, Resources | Marjan van Esbroeck, Methodology, Project administration, Resources, Supervision | Kristien Verdonck, Investigation, Methodology, Project administration, Resources | Marleen Boelaert, Methodology, Project administration, Resources | Nitin Gupta, Methodology, Validation | Pierre-Édouard Fournier, Methodology, Project administration, Resources, Supervision | Emmanuel Bottieau, Project administration, Resources, Supervision, Validation

## ETHICS APPROVAL

The NIDIAG study was initially registered at clinicaltrials.gov under the identifier NCT01766830 and was conducted in line with a specifically designed ethics Charter (www.NIDIAG.eu). The study was initially approved by the Institutional Review Board of the ITM under the number 12125818 (June 5, 2012), with a granted amendment for additional testing number 793/11 (January 30, 2024). The study was also approved by the Ethics Committee of the Antwerp University Hospital/ University of Antwerp (UZA/ UA) under the Belgium registration number B300201214571 (June 25, 2012) with granted approval for additional testing "Project ID 6871 - Edge n/a - BUN: B3002024000144" (August 12, 2024). The study was approved by ethical committees of partner institutions in the countries where data were collected: The Ethics Committee of the University of Khartoum (Khartoum, Sudan, FEM/DO/EC102; Submission: June 3, 2012; Approval: June 13, 2012; Update: January 18, 2024), the National Research Ethics Review Committee, Sudan (Khartoum, Sudan; Submission: September 5, 2012; Approval: November 18, 2012; Update: January 21, 2024), the Ethics Committee of B.P. Koirala Institute of Health Sciences (BPKIHS, Dharan, Nepal, 57/2012; Submission: May 29, 2012; Approval: July 3, 2012; Update: January 18, 2024), National Ethics Committee for Health Research of Cambodia (Phnom Penh, Cambodia, 134/2012; Submission: May 27, 2012; Approval: August 24, 2012; Update: January 20, 2024) and the University of Kinshasa School of Public Health (Kinshasa, Democratic Republic of the Congo, ESP/CE/016; Submission: December 4, 2012; Approval: January 14, 2013; Update: January 19, 2024). All study participants or their guardians gave written informed consent, and children <18 years also gave their assent. The original informed consent included a statement about sample storage for future research on infectious diseases. Samples from participants who opted for post-study destruction at the conclusion of the original NIDIAG protocol were excluded from biobanking. For further details, please see: https://nidiag.eu/.

## ADDITIONAL FILES

The following material is available online.

## Open Peer Review

**PEER REVIEW HISTORY (review-history.pdf).** An accounting of the reviewer comments and feedback.

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
