## [Reviewer comments · Microbiology Spectrum]

Microbiology Spectrum

***Coxiella burnetii* and *Bartonella* species serology of febrile patients with an established infectious or inflammatory diagnosis in Sudan, Nepal and Cambodia**

Carl Boodman, Sophie Edouard, Johan van Griensven, Kanika Koirala, Basudha Khanal, Suman Rijal, Narayan Bhattarai, Sayda Safi, Thong Phe, Kruy Lim, Pascal Lutumba, François Chappuis, Cédric Yansouni, Achilleas Tsoumanis, Barbara Barbé, Marjan Van Esbroeck, Kristien Verdonck, Marleen Boelaert, Nitin Gupta, Pierre-Edouard Fournier, and Emmanuel Bottieau

Corresponding Author(s): Carl Boodman, Instituut voor Tropische Geneeskunde

Review Timeline:

Submission Date:	June 17, 2025
Editorial Decision:	July 19, 2025
Revision Received:	July 20, 2025
Editorial Decision:	August 1, 2025
Revision Received:	August 4, 2025
Accepted:	August 22, 2025

Editor: John Atack

Reviewer(s): The reviewers have opted to remain anonymous.

Transaction Report:

DOI: <https://doi.org/10.1128/spectrum.01675-25>

Re: Spectrum01675-25 (*Coxiella burnetii* and *Bartonella* species serology of febrile patients with an established infectious or inflammatory diagnosis in Sudan, Nepal and Cambodia)

Dear Dr. Carl Boodman:

Thank you for the privilege of reviewing your work. Below you will find my comments, instructions from the Spectrum editorial office, and the reviewer comments.

Please resubmit this article via the Spectrum submission system in order to complete the spectrum submission form before we can accept this article. I am happy with the response to reviewers from your previous submission, and this is just an editorial office requirement

Revision Guidelines

Sincerely,
John Attack
Editor
Microbiology Spectrum

Re: Spectrum01675-25R1 (*Coxiella burnetii* and *Bartonella* species serology of febrile patients with an established infectious or inflammatory diagnosis in Sudan, Nepal and Cambodia)

Dear Dr. Carl Boodman:

Thank you for the privilege of reviewing your work. Below you will find my comments.

Please include IRB/Ethics committee approval numbers from all participating institutions within the ethics section. I appreciate the statement has been modified in response to a comment from reviewer #2. However, as samples relate to human subjects, it is necessary to include all relevant data/reference numbers in the manuscript and not just cite a reference/link. Please modify the ethics statement accordingly to include all relevant IBC approval numbers from all institutions, even if this has been cited before in previous publications.

Revision Guidelines

Sincerely,
John Attack
Editor
Microbiology Spectrum

Response:

Please include IRB/Ethics committee approval numbers from all participating institutions within the ethics section. I appreciate the statement has been modified in response to a comment from reviewer #2. However, as samples relate to human subjects, it is necessary to include all relevant data/reference numbers in the manuscript and not just cite a reference/link. Please modify the ethics statement accordingly to include all relevant IBC approval numbers from all institutions, even if this has been cited before in previous publications.

Response: Thank you. This has been done. We were thankfully able to access the NIDIAG archives at the Institute of Tropical Medicine in Belgium. Some of these documents are no longer accessible in the original countries. Please see correspondence with Prof. Sayda El Safi from Sudan as an example:

8/4/25, 5:35 PM

Mail - Carl Boodman - Outlook

Dear Sayda,

I hope you're doing well and thank you again for your participation in the NIDIAG fever projects and recent amendments with Bartonella and Coxiella testing.

For our second paper, the last remaining issue that editors would like is to include the original IRB numbers from the participating sites. For some reason, this wasn't referenced in the other articles in the past and we're having some trouble locating them as Prof. Boelaert died.

Any chance you happen to have an IRB number we can quote?

Thanks again and all the best,

Carl

Carl Boodman (he/ him), MD, FRCPC, DTM&H, CTropMed®
Infectious Diseases and Medical Microbiology
PhD Candidate, University of Antwerp/ Institute of Tropical Medicine (Belgium)
Clinical Investigator Program Candidate, University of Manitoba (Canada)
Member, International Diagnostics Centre
boodmanc@myumanitoba.ca

From: Sayda El Safi <shelsafi@yahoo.com>

Sent: Sunday, August 3, 2025 4:30 PM

To: Carl Boodman <boodmanc@myumanitoba.ca>

Subject: Re: Original NIDIAG IRB Number Sudan 2012

Caution! This message was sent from outside the University of Manitoba.

Dear Carl,

I am very sorry for not being able to provide you with this information.

I have been outside Sudan since the start of the war in my country in April 2023 when we had to leave our home in less than thirty minutes, leaving behind all our belongings.

I hope the reviewer would consider these unavoidable circumstances.

I wish you all the best with your submission.

Sayda

Sent from Yahoo Mail for iPhone

On Saturday, August 2, 2025, 4:08 pm, Carl Boodman <boodmanc@myumanitoba.ca> wrote:

Re: Spectrum01675-25R2 (*Coxiella burnetii* and *Bartonella* species serology of febrile patients with an established infectious or inflammatory diagnosis in Sudan, Nepal and Cambodia)

Dear Dr. Carl Boodman:

Your manuscript has been accepted, and I am forwarding it to the ASM production staff for publication. Your paper will first be checked to make sure all elements meet the technical requirements. ASM staff will contact you if anything needs to be revised before copyediting and production can begin. Otherwise, you will be notified when your proofs are ready to be viewed.

Sincerely,
John Attack
Editor
Microbiology Spectrum